# The Reliability of Field-Based Static and Dynamic Balance Tests in Primary School-Aged Autistic Children

**DOI:** 10.3390/bs14080640

**Published:** 2024-07-25

**Authors:** Emma Baldwin, Sharon Kinsella, Paul J. Byrne

**Affiliations:** Department of Health and Sport Sciences, South East Technological University, Kilkenny Road Campus, R93 V960 Carlow, Ireland; c00224273@setu.ie (E.B.); sharon.kinsella@setu.ie (S.K.)

**Keywords:** autism, test–retest, stability, coefficient of variance

## Abstract

Previous research has proven that the balance of autistic children is poor. However, the reliability of assessing balance in this cohort has been inadequately researched. This study therefore aimed to examine if field-based static and dynamic balance tests can be reliably assessed in autistic children, to determine the number of familiarisation sessions required and whether autistic severity impacts on the reliability of these balance tests. The balance of eighteen primary school-aged autistic children was assessed three times a week over five weeks, using the flamingo balance test, a modified version of the balance error scoring system (BESS), the low beam walking test, and the heel to toe walking test. Reliability criteria included an intraclass correlation coefficient (ICC) level of ≥0.75 and a coefficient of variance (CV%) of ≤46% for the low beam walking test, the heel to toe walking test, and the BESS, and a CV% of ≤82% or the flamingo balance test. Inter-session reliability was achieved and required the least number of familiarisation sessions for the flamingo balance test, compared to the low beam walking test, which required a greater number of familiarisation sessions to achieve inter-session reliability. The heel to toe walking test and the BESS achieved inter-session reliability and familiarisation in an acceptable time frame. Due to the large CV% values reported in the current study, practitioners need to be aware that balance interventions need to achieve improvements greater than the CV% in this cohort.

## 1. Introduction

Autism is a spectrum of developmental conditions which can significantly impact on an individual’s communication, social interaction, and behaviours [1]. The prevalence of autism has been increasing dramatically in recent years, and can vary in severity from mild to moderate to severe [2]. Worldwide, it is estimated that approximately one in one hundred children has autism [3]. This figure is much higher in the United States of America (USA), where it is estimated that approximately one in thirty-six 8-year-old children has autism [4]. Autism has been more commonly reported in males with a 3:1 ratio [5]. Nearly 75% of autistic children in the USA have co-occurring conditions such as attention-deficit hyperactivity disorder (ADHD), anxiety, depression, sensory problems, non-verbal learning disorders, obsessive compulsive disorder (OCD), immune disorders, and sleep disorders [6]. Additionally, it is estimated that approximately 30–40% of autistic children have an intellectual disability [7]. Autistic children are also less active than their neurotypically developing peers [8], due to interpersonal, intrapersonal, and environmental factors that impact their exercise participation [9], and therefore are at a higher risk of developing diabetes, stroke, obesity, heart disease, and depression [10]. Many studies have shown that autistic children perform more poorly across many fitness parameters including strength, flexibility, and, in particular, balance when compared to neurotypically developing children [11,12].

Balance can be defined as “the maintenance of equilibrium while stationary or moving” [13]. Balance is a complex task which is multidimensional in nature and generally is an unconscious process in the human body [14]. Balance can be subdivided into two classes: static balance, which requires maintaining equilibrium whilst standing in one spot (e.g., flamingo balance test), or dynamic balance, which requires the maintenance of equilibrium whilst moving (e.g., low beam walking test) [14]. Balance is determined by the position of the centre of mass (COM) and the area of base of support (BOS), and therefore for an object to be balanced, the line of gravity needs to fall within the BOS of the object [15]. Conversely, the object will become unbalanced if the line of gravity falls outside of the BOS [15]. In humans, there are many factors that can affect an individual’s balance, including the size of BOS, the height of the centre of gravity (COG), and the line of gravity in relation to the BOS [16]. To maintain balance, our bodies need interaction from our musculoskeletal, neural, visual, vestibular, and somatosensory systems [17]. The sensory processing experienced from these systems travels to our central nervous system, generates a balance response, and allows us to sustain balance [18]. Those with autism often have sensory processing deficits, and so information processing to and from the brain is affected, especially in relation to the planning, timing, and completion of movements [19]. Autistic children have poorer balance due to changes in both the structure and function of the cerebellum and the basal ganglia [20].

The existing literature attributes more proficient fundamental movement skills such as running, hopping, jumping, catching, and throwing, to those who possess better balance [21]. However, many school-aged autistic children find these gross motor skills to be challenging [22]. In addition, those who possess poor postural stability are more likely to possess social communication issues, more repetitive behaviours, and more challenges emotionally and behaviourally, and to have a lower IQ [12,23,24,25]. It is therefore paramount to improve balance in this cohort. There are numerous studies that have shown the large positive effects of physical activity interventions on balance in autistic children [26,27]. However, the reliability of assessing both static and dynamic balance in this cohort has been inadequately researched. Previous research used a variety of different balance tests, some with reliability values attributed to autistic children, but most without. Previously, single-leg balance tests have been used to assess static balance in autistic children such as the flamingo balance test or the stork balance test [27,28]. However, no reliability values were reported for the flamingo balance test in autistic children [26]. ICC values of 0.90 and a CV% value of 26.7% were reported in a study where the stork balance test was used to assess static balance in ninety-two 4–13-year-old autistic children [28]. Additionally, static balance is commonly assessed using a variety of conditions such as double-leg, single-leg, and tandem standing with variations of eyes open or eyes closed such as the balance error scoring system (BESS) or the NeuroCom Balance Master. Both have been used to assess balance in autistic children; however, no reliability values were reported [29,30].

Another common battery of tests used when assessing balance in autistic children is the Bruininks–Oseretsky Test of Motor Proficiency (BOTMP) or BOTMP-2, which consists of nine balance tests which assess both static and dynamic balance [31]. Interrater reliability for the BOTMP test ranges from 0.38 to 0.92 and for the BOTMP-2 interrater reliability ranges from 0.6 to 0.92 for children and youth up to twenty-one years old [31]. Both the heel to toe walking test and the low beam walking test, which are components of the BOTMP test, have been previously used to test balance in autistic children; however, there is a lack of reliability studies for this cohort [11,32,33,34,35,36,37,38,39]. Similarly, the heel to toe walking test is also included in the balance assessment for the Movement Assessment Battery for Children (MABC-2). The MABC-2 is a widely recognised tool for assessing balance in children with mild to moderate motor impairments [40] and has been used to assess balance in autistic children [33].

Though many of these studies do not focus on balance training specifically as an intervention, these studies showed significant improvements in balance in autistic children through the practice of non-balance-focused physical activity [11,26,33,34,35,36,37,38,39]. Additionally, balance performance has been shown to be affected by autism severity [12,24,41], where those with more severe autistic symptoms had poorer balance and more postural sway during static balance [12,41]. However, although many intervention studies have utilised many different methods of assessing autism severity, few report results in relation to severity level [11,27,33,42].

It is recommended that autistic children undergo a familiarisation or practice period to minimise the learning effect. It was established that for full engagement from autistic children, four weeks of familiarisation are needed following an integrative neuromuscular training intervention [43]. Other previous research assessing balance intervention programmes in autistic children has not stated whether familiarisation for test protocols was completed. Additionally, familiarity and routine are crucial to autistic children, which comes with practice and prevents any unnecessary anxiety for the children [44].

As a result of the existing literature highlighting the extreme balance difficulties faced by autistic children, and the lack of reliability data available in the literature surrounding balance assessment methods for autistic children, the current study aims (1) to assess the reliability of field-based static and dynamic balance tests in primary school-aged autistic children using the low beam walking test, the heel to toe walking test, the flamingo balance test, and a modified version of the BESS; (2) to determine the number of familiarisation sessions required to achieve reliable balance tests in primary school-aged autistic children; and (3) to examine the impact of autistic severity levels on the reliability of balance tests in primary school-aged autistic children.

## 2. Materials and Methods

### 2.1. Participants

Forty-six children who were educated in a specialised educational setting, within mainstream primary schools, were recruited to participate voluntarily in this study. Of the forty-six children who consented, forty-four children participated in the programme; however, only eighteen autistic children (thirteen boys and five girls) completed 80% of the sessions (12 out of 15 sessions) and were included in the final analysis (mean age 9.22 years ± 1.59) [Table 1]. Inclusion criteria for autistic children included (a) previously having a formal diagnosis of autism by an educational psychologist; (b) being accepted by the Department of Education into a specialised educational autism setting within a mainstream primary school; and (c) being aged 4–13 years old. Children were not included in the study if they had no formal diagnosis of autism, or if they had any injuries or illnesses which prevented them from participating in physical activity. Written consent was obtained from the schools’ principal, the child’s teachers, and the child’s parents/ guardians, along with participant assent prior to the study commencing, and all participants were medically screened. Ethical approval was obtained from the South East Technological University’s ethics committee (Ethics Approval Number: 304). Teachers were asked to complete the Gilliam Autism Rating Scale (GARS) 2nd Edition [45] which has been proven to be both valid and reliable [46], in order to classify the autism severity of the participants into one of three categories: mild (n = 8), moderate (n = 10), or severe (n = 0).

### 2.2. Procedures

All testing sessions occurred at the same time of day in each school on a Monday, Wednesday, and Friday, for five consecutive weeks (15 sessions) indoors. Sessions varied from 30 min to 1 h depending on the number of children per session. Children were unfamiliar with the balance tests prior to the study commencing. The children completed a warm-up at the start of every session. The children removed their shoes for the duration of the tests. The participants were provided with both verbal and visual instruction from the instructor and using The Story Creator Application (Innovative Mobile Apps Ltd., Woodstock, GA, USA) displayed on an iPad (iPad Air 2, Model A1566, Apple Inc., Cupertino, CA, USA) which broke down the tests using key pictures and key words and was accessible throughout the session.

Flamingo Balance Test: The children were required to stand on their dominant leg (determined by the leg they kick a ball when rolled at them) on a low beam (150 mm high, 300 mm wide) [Beemat Foldable Balance Beam, Beemat, Dewsbury, UK]. The foot of the non-dominant leg was held with the hand on the same side, whilst the leg was flexed at the knee for one minute, during which the stopwatch was stopped if the participant lost balance by falling off the beam or released the foot. The stopwatch was re-commenced when the participant reassumed the testing position and the number of falls was recorded [47].

BESS: Participants stood with their hands on their hips with their eyes closed on a firm surface for twenty seconds in a double-leg stance, a single-leg stance (on their non-dominant foot), and a tandem stance (non-dominant foot in the back). Errors were counted and included participants opening their eyes, moving their hands off their hips, stumbling or falling, lifting their heel or forefoot off the floor, and being out of the testing position for more the five seconds. Only one error was counted if multiple occurred at the same time. The maximum number of errors in one stance was ten and the participants were scored out of thirty [48].

Low Beam Walking Test: Participants were required to walk the length of a low beam (150 mm high, 300 mm wide, 4 m long) as rapidly as possible. The participants started in front of the beam and were timed in seconds to walk from one end of the beam to the other, whilst keeping their hands on their hips [33,36,42,47].

Walking Forward Heel to Toe Along a Line Test: Participants were required to walk heel to toe along a line for fifteen steps. The test was stopped if the participant deviated off the line, and the test was scored based upon the number of steps counted [11,33,34,35,36,37,38,39].

### 2.3. Statistical Analysis

All statistical analyses were conducted using IBM Statistical Package for Social Sciences (SPSS, version 28, Armonk, New York, NY, USA). All data were screened for normality using the Shapiro–Wilk test. Inter-session reliability was assessed using an intraclass correlation coefficient (ICC) [two-way mixed model for absolute agreement, average measure] and a coefficient of variance (CV%) with 95% confidence interval (CI). ICC thresholds were used to deem the reliability of the tests as either poor (ICC > 0.5), moderate (ICC 0.5–0.75), good (ICC 0.75– 0.9), or excellent (ICC <0.9) [49]. The criteria which deemed reliability for a balance test included an ICC level ≥0.75 and a CV% which was equal to or below the thresholds which are set out below.

The CV%s that were deemed acceptable were based on a judgement made by the lead author. These CV%s were judged to be acceptable based upon a meta-analysis and percentage improvements following interventions in previous studies [11,26,27,30,35,36,37,38,39,50]. The meta-analysis showed that the average Hedges’ g following a balance intervention programme in autistic youth was 1.82 [27]. For the low beam walking test, the heel to toe walking test, and the BESS test, a CV% threshold of ≤46% was deemed acceptable based upon previous physical activity intervention programme findings in relation to percentage change and effect size (Hedges’ g). For the low beam walking test, a previous study found a 151.11% (Hedges’ g = 1.65) improvement [50]; the heel to toe walking test has been found to improve by 33.33–417.33% (Hedges’ g = 0.99–4.72) [11,27,32,33,34,35] and the BESS test has shown a 6.35–52.44% (Hedges’ g = 0.51) improvement using the Neurocom Balance Master, which is a similar test to the BESS [30]. For the flamingo balance test, a CV% threshold of ≤82% was set, based upon previous studies showing a 39.88–111.97% (Hedges’ g = 1.72–1.87) improvement in balance following a physical activity intervention programme [26,27,50].

The number of familiarisation sessions required for the balance tests was determined by identifying where there was no significant change between sessions prior to a session being deemed reliable based upon the criteria above. A repeated measures analysis of variance (ANOVA) or Friedman’s test with post hoc analysis was used to identify significant differences between sessions. An alpha level of *p* < 0.05 was used for all analyses.

## 3. Results

The results of this study are displayed in Table 2. A summary of results can also be found in Figure 1.

### 3.1. Low Beam Walking Test

Based on the reliability criteria for this study, inter-session reliability was achieved on session 8 (ICC = 0.80; CV% = 40.49%) for this test. This test required seven familiarisation sessions to achieve reliable data. The mild severity group achieved reliability on session 3 (ICC = 0.92; CV% = 41.51%), compared to session 5 in the moderate severity group (ICC = 0.79; CV% = 34.37%). The mild severity group required two sessions to achieve familiarisation, in comparison to the moderate severity group who required four sessions (Table 2).

### 3.2. Walking Forward Heel to Toe Along a Line Test

Inter-session reliability was achieved on session 4 for this test, based on the reliability criteria for this study (ICC = 0.89; CV% = 45.73%). This test required three familiarisation sessions. The mild severity group achieved reliability on session 2 (ICC = 0.81; CV% = 21.87%), compared to session 4 in the moderate severity group (ICC = 0.93; CV% = 45.87%). The mild severity group required one session to achieve familiarisation, in comparison to the moderate severity group who required three sessions (Table 2).

### 3.3. Flamingo Balance Test

Inter-session reliability was achieved on session 2 for this test (ICC = 0.77; CV% = 81.18%). The flamingo balance test required only one familiarisation session. The mild severity group achieved reliability on session 9 (ICC = 0.91; CV% = 70.62%) compared to session 4 in the moderate severity group (ICC = 0.95; CV% = 69.69%). The mild severity group required eight sessions to achieve familiarisation, in comparison to the moderate severity group who required three sessions (Table 2).

### 3.4. BESS

Inter-session reliability was achieved on session 5 for this test, based on the reliability criteria for the current study (ICC = 0.90; CV% = 43.04%). This test required four sessions for familiarisation to occur. The mild severity group achieved reliability on session 8 (ICC = 0.79; CV% = 41.72%) compared to session 5 in the moderate severity group (ICC = 0.89; CV% = 37.09%). The mild severity group required seven sessions to achieve familiarisation, in comparison to the moderate severity group who required four sessions (Table 2).

## 4. Discussion

The aim of the current study was to firstly examine if the low beam walking test, the heel to toe walking test, the flamingo balance test, and the BESS can reliably assess static and dynamic balance in primary school-aged autistic children. All these tests were found to be reliable; however, the number of sessions required to achieve reliability varied for each test. The flamingo balance test achieved inter-session reliability on session 2 (ICC = 0.77; CV% = 81.18%), followed by the heel to toe walking test, where inter-session reliability was achieved on session 4 (ICC = 0.89; CV% = 45.73%), and the BESS where inter-session reliability was achieved on session 5 (ICC = 0.90; CV% = 43.04%). The low beam walking test achieved inter-session reliability on session 8 (ICC = 0.80; CV% = 40.49%). The second aim of the current study was to determine the number of familiarisation sessions required to achieve reliable balance tests in primary school-aged autistic children. The low beam walking test required the largest amount of familiarisation sessions at seven, followed by the BESS which required four sessions, the heel to toe walking test that required three familiarisation sessions, and the flamingo balance test that only required one familiarisation session. The third aim of the current study was to examine the impact of autistic severity levels on the reliability of balance test in primary school-aged autistic children. It appears that the mild severity group familiarised and achieved inter-session reliability faster for the dynamic balance tests (i.e., the low beam walking test and the heel to toe walking test), compared to the moderate severity group, who required fewer familiarisation sessions and achieved inter-session reliability earlier for the static balance tests (i.e., the flamingo balance test and the BESS).

The reliability of the heel to toe walking test was difficult to compare to previous reported results as it has been used as part of a larger battery of tests, the BOTMP or BOMTP-2, in which the reliability values were often reported for the balance section as a whole, rather than the individual balance components. The BOTMP and BOTMP-2 report reliability values for the balance portion of the battery of tests in autistic children and range from 0.63 to 0.97 for inter-rater reliability and 0.38 to 0.92 for internal consistency [27]. However, the heel to toe walking test, which is also part of the Movement Assessment Battery for Children, had a much smaller range of ICC values of 0.78–0.86 in autistic children with movement disorders and compares favourably to the current study [33]. The heel to toe walking test was also used to test dynamic balance in fifty-six 8–14-year-old autistic children; however, no reliability values were reported [11]. Similar to the heel to toe walking test, the low beam walking test is also a component of the BOTMP and BOTMP-2, which, as mentioned above, have varied reliability. In addition to the large range of ICC values reported for the BOTMP and BOTMP-2 [22], the low beam walking test in the current study had comparable ICC values (ICC = −0.22–0.80) [Table 2].

In relation to the static balance test, the flamingo balance test in the current study had large ranges of ICC and CV% values (ICC = 0.52–0.87; CV% = 59.17–117.44%) [Table 2], which is in contrast to the ICC values of 0.90 and CV% of 26.7% reported when using the stork balance test to assess static balance in autistic children [28]. In addition, the flamingo balance test in the current study is consistent with the large range of ICC values, namely, 0.40–0.80, reported when assessing balance in autistic children [27]. Similarly, large ICC and CV% ranges were found in this study for the BESS (ICC = −0.05–0.96; CV% = 23.97–72.07%) [Table 2]. Excellent test–retest reliability values of the BESS were also reported, as well as interrater ICC values of 0.93 and intra-rater ICC values of 0.96, in three hundred and seventy-three 5–14-year-old neurotypically developing children [51].

It is difficult to compare familiarisation results with previous research as most studies did not state whether familiarisation for test protocols was completed. However, four weeks of familiarisation were required for full engagement from autistic children, following an integrative neuromuscular training intervention [43]. In the current study, the low beam walking test required the most familiarisation sessions, namely, seven (Table 2). This test would therefore not be recommended by the authors due to the feasibility of requiring so many familiarisation sessions for a balance test. Conversely, the flamingo balance test required only one familiarisation session to produce reliable results; however, it would be advisable to have two to three familiarisation sessions to reduce the variance in this cohort. Similarly, the heel to toe walking test only required three sessions for familiarisation to occur, whilst the BESS required four sessions to produce reliable results and have a minimal learning effect.

From a severity level perspective, it appears that the mild severity group became familiar with the test earlier than the moderate severity group in the dynamic balance tests, as they required fewer familiarisation sessions to produce reliable results and produced reliable results at much earlier time points. Similarly, for the dynamic balance test, it appears that the mild severity group had a lower CV% than the moderate severity group. Conversely, the moderate severity group achieved familiarisation earlier in the static balance tests than the mild severity group. The moderate severity group required fewer familiarisation sessions to produce reliable results and had a lower CV%. Neither the BESS nor the flamingo balance test would be recommended for a mild severity group based on the results of this study, due to the feasibility of requiring seven and eight familiarisation sessions, respectively. Balance performance has been shown to be affected by autism severity [28], with those with more severe autistic symptoms generally having poorer balance [12,41]. However, none of the children assessed in the current study had severe autism symptoms, and thus it is unclear how severe autistic symptoms would affect the reliability of the balance tests used and how familiarisation would be affected.

In the current study, the CV%s for the four balance tests are high, ranging from 21.87% to 81.18%. A judgement was made whereby an acceptable CV% for the low beam walking test, the heel to toe walking test, and the BESS test was set at ≤46%. Moreover, the flamingo balance test had an acceptable CV% set at ≤82%. Justifications for these accepted CV% limits are provided in the statistical analysis part of the Materials and Methods section. In relation to previous work, a high CV% of 26.7% was calculated for the stork balance test when it was used to assess balance in primary school-aged autistic children [28]. Moreover, when other variables were tested in the same cohort, such as the sit and reach test to assess flexibility, the CV% was 2% [28]. This previous study provides some evidence that autistic children have higher CV%s than neurotypically developing children and the CV% in balance tests appears higher than other physical fitness tests such as the sit and reach test. The high CV%s in balance in autistic children may be a result of neurophysiological, environmental, or behavioural factors. From a neurological perspective, autistic children have been associated with reduced cerebellar activation, cerebellar undergrowth, and changes in both the structure and function of the basal ganglia and cerebellum, all of which have been associated with postural control issues [20,52]. This may account for differences in CV%s between balance in autistic children and neurotypically developing children. In addition, due to the nature of autism, an autistic child having a negative behavioural episode during a balance session may lead to other autistic children also participating in the session to experience anxiety, stress, and upset [9], and ultimately leads to the children been distracted and performing below the best of their abilities, which may lead to high CV%s. Furthermore, environmental factors such as the equipment (force plates in particular) may have impacted hyper or hypo sensitivities experienced by the autistic children and ultimately affected the variance and the reliability of the balance tests over the fifteen sessions.

### 4.1. Practical Applications

This study provides much needed reliability data for the low beam walking test, the heel to toe walking test, the flamingo balance test, and the BESS for primary school-aged autistic children. In addition, it is important that researchers note that a number of familiarisation sessions are needed by autistic children to become familiar with the balance tests prior to collecting balance. Figure 1 outlines the key points for each balance test used in this study and could be a useful tool for future practitioners to refer to when selecting a balance test for their cohort. Future practitioners would benefit from conducting a screening process prior to balance testing with autistic children to identify key information. Key information could include easy versus hard balance tests, which were favoured or unfavoured by the autistic children, which test suits mild severity versus moderate severity groups better, and whether the equipment suits their sensory needs. This process may aid future practitioners in choosing the most appropriate balance test for a group of autistic children to achieve the most reliable results. By establishing this key information, future researchers may be able to decipher whether an appropriate amount of enjoyment coupled with an appropriate amount of challenge aids in producing the most reliable results in this cohort in terms of balance. This ‘Goldilocks Principle’ or ‘just right’ paradigm [53] was observed in the current study, where it appeared to the researchers that the flamingo balance test was difficult, but was also enjoyed and achieved inter-session reliability the quickest. In contrast, the low beam walking test achieved inter-session reliability after the longest period of time, even though it appeared to the researchers to be the easiest test and was enjoyed by the children. However, enjoyment and difficulty were not objectively measured in this study; they were observed by the researchers, and hence objectively measuring these factors could be an avenue for future researchers to establish.

Secondly, due to the large CV% values seen across this study, practitioners should be aware, when conducting future interventions on balance in this cohort, that the improvements they need to achieve must be greater than the CV% noted in this study.

### 4.2. Limitations and Future Research

There are numerous limitations to this study that should be considered. Firstly, the CV% values were high across a large proportion of the test sessions, thus showing large variability in this cohort. Researchers should therefore be aware when conducting balance interventions with this cohort that the improvements they need to achieve must be greater than the large CV%s observed in this study. This study, however, is an important addition to the limited research available on balance in autistic children and adds much needed reliability values, both in terms of ICC and CV%.

Secondly, due to the nature of the primary school’s system in Ireland, school holidays, days off from school, and bank holidays, it was difficult to keep all the different groups of children on the same pattern of Monday, Wednesday, and Friday consecutively for 5 weeks without missing days. This made it difficult to compare results on a session-to-session basis at the end of the testing period. It also made it difficult to assess if the reliability of the autistic children’s balance varied depending on the day of the week, and perhaps this is an area that future research could address.

Lastly, only a small proportion of the sample of participants were females (5, compared to 13 boys). This causes difficulty in applying these findings to autistic girls and hence future studies should include a more balanced split between male and female participants. Similarly, future research should aim to include children with more severe autistic symptomology. This was a limitation for the present study as to be included in the final data analysis, participation in all four balance tests was necessary. Similarly, the large age range of autistic children in the small sample size may have impacted the reliability and variability of the balance tests in the current study, and thus further research is necessary to address the extent at which age may affect the reliability of measuring balance in autistic children.

## 5. Conclusions

In the current study, the reliability of four balance tests in primary school-aged autistic children three times per week over a 5-week period was examined. All these tests were found to be reliable; however, the number of sessions required to achieve reliability varied for each test. The flamingo balance test achieved inter-session reliability with the least number of familiarisation sessions required, despite appearing difficult to the autistic children. Conversely, the low beam walking test took the longest to achieve inter-session reliability, and required an impractical amount of familiarisation sessions, even though it appeared to be enjoyed by the children. Despite appearing to be disliked by the autistic children, the heel to toe walking test and the BESS achieved inter-session reliability and familiarisation in an acceptable time frame. Additionally, the mild severity group required fewer familiarisation sessions and achieved inter-session reliability more quickly for the dynamic balance tests, whilst the moderate severity group required fewer familiarisation sessions and achieved inter-session reliability more quickly for the static balance tests.

Overall, there are many benefits of improved balance for autistic children, and balance interventions and exercises should be incorporated into the lives of these children from a young age to reduce the balance deficits that they frequently display.

## Figures and Tables

**Figure 1 behavsci-14-00640-f001:**
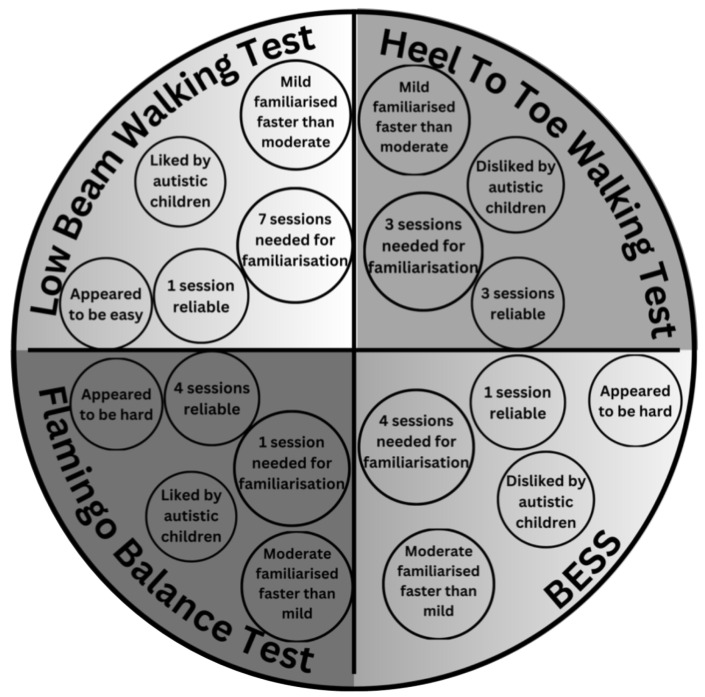
Key points for the four tests.

**Table 1 behavsci-14-00640-t001:** Breakdown of participants who were included in the final analysis of results.

Age	Severity	Gender
7	Mild (n = 2) Moderate (n = 1)	Male (n = 2) Female (n = 1)
8	Mild (n = 2) Moderate (n = 2)	Male (n = 4) Female (n = 0)
9	Mild (n = 0) Moderate (n = 2)	Male (n = 2) Female (n = 0)
10	Mild (n = 1) Moderate (n = 2)	Male (n = 2) Female (n = 1)
11	Mild (n = 2) Moderate (n = 2)	Male (n = 2) Female (n = 2)
12	Mild (n = 0) Moderate (n = 1)	Male (n = 1) Female (n = 0)

**Table 2 behavsci-14-00640-t002:** Summary of results from the four balance tests across the fifteen sessions for the overall group, and for the mild and moderate severity groups.

	ICC Range	Reliable Session ICC	CV% Range	Reliable Session CV%	Reliable Session No.	Familiarisation Sessions Required
LBWT (Overall)	(−0.22–0.80)	0.80	(30.87–59.38%)	40.49%	8	7
LBWT (Mild)	(−0.62–0.92)	0.92	(29.80–55%)	41.51%	3	2
LBWT (Moderate)	(−0.73–0.86)	0.79	(28.61–68.71%)	34.37%	5	4
HTWT (Overall)	(−0.33–0.89)	0.89	(6.24–54.83%)	45.73%	4	3
HTWT (Mild)	(−0.13–0.92)	0.81	(2.52–53.94%)	21.87%	2	1
HTWT (Moderate)	(−0.35–0.93)	0.93	(6.97–58.08%)	45.87%	4	3
FBT (Overall)	(0.52–0.87)	0.77	(59.17–117.44%)	81.18%	2	1
FBT (Mild)	(−0.03–0.91)	0.91	(41.26–111.67%)	70.62%	9	8
FBT (Moderate)	(0.59–0.95)	0.95	(62.76–133.50%)	69.69%	4	3
BESS (Overall)	(−0.05–0.96)	0.90	(23.97–72.07%)	43.04%	5	4
BESS (Mild)	(−0.76–0.98)	0.81	(32.89–77.18%)	41.72%	8	7
BESS (Moderate)	(0.06–0.94)	0.89	(27.78–72.84%)	37.09%	5	4

BESS = Balance Error Scoring System; FBT = Flamingo Balance Test; HTWT = Heel to Toe Walking Test; LBWT = Low Beam Walking Test.

## Data Availability

The data supporting the conclusions of this article will be made available by the authors on request.

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
