# Peer review of "The Reliability of Field-Based Static and Dynamic Balance Tests in Primary School-Aged Autistic Children"

_behavsci, 2024, doi:10.3390/bs14080640_

Round 1

Reviewer 1 Report

Comments and Suggestions for Authors

The paper entitled “The Reliability of Field Based Static and Dynamic Balance Tests in Primary School Ages Autistic Children” aims to assess (1) reliability of field-based static and dynamic balance tests in children with autism; (2) number of familiarization sessions required for each test to achieve reliability; (3) impact of autism severity on tests’ outcomes. Overall the paper is very interesting and addresses a relevant theme for research on gross motor skills and balance in Autism Spectrum Disorders (ASD). However, I think the paper in its current format requires some revision before being accepted for publication. Here are some comments:

1. I think the main limitation to this work is the limited number of participants, also considering the wide age range (from 4 to 13 years of age. Recruitment included 46 children with ASD, but the final sample was reduced to 18 children. I fully understand difficulties outlined by the authors (also due to school holidays in Ireland), but I suggest that they:

- add a more detailed description of the participants’ characteristics describing not only mean age and standard deviation in the text, but providing a participants table with indications on the actual age range and a more detailed description of autism severity in their sample.

- describe how the formal diagnosis was carried out (was the ADOS used? Can the ADOS scores be added to the paper or the participant table?)

- list among the limitations of the current study the small sample size (and, eventually, also the wide age range considered). In fact, this may not only explain the “large variability” that they found in their data, but also provide some further indications to readers on how/to what extent age may impact performance in balance tests.

2. multiple studies on gross motor skills and balance in children with autism have relied on the Movement Assessment Battery for Children (or Movement ABC), which also allows assessing fine motor skills. This tool is cited by the Authors in the Discussion, but I believe it would be useful for readers if it could be mentioned and discussed (possibly including references to dedicated literature) also in the Discussion. I think that providing an indication of the relation between this widely used standardised tool and the less know, but surely effective, balance tests that are the object of the present study, would be useful for readers.

3. In Figure 1 Authors report whether tests were liked/disliked by autistic children. However, I could not find any indication in the text of how this was measured. It is relevant and useful information, but it would be even more useful if Authors described how it was obtained (even if it was only based on field observations it is important to say so).

Reviewer 2 Report

Comments and Suggestions for Authors It is a good work and is very well structured, with a solid statistical analysis that gives consistency to the results and conclusions.
I suggest giving a name to the battery of tests selected for this study on the evaluation of static and dynamic balance in autistic children. This gives greater strength to the study since methodologically it is very consistent.
Likewise, I recommend that the data be clarified according to the severity of the type of autism and the relationship with the balance evaluated, that is, give greater specificity about the reliability, especially with the most severe ones.  

Reviewer 3 Report

Comments and Suggestions for Authors

Authors should begin by presenting a definition of intellectual disability, stating that autism is a subgroup.

The APA reference should be updated.

Why are the authors presenting information about Ireland?

Why is this population less active? Is it because there are barriers to physical activity? I suggest the following reading: Jacinto, M., Vitorino, A. S., Palmeira, D., Antunes, R., Matos, R., Ferreira, J. P., & Bento, T. (2021). Perceived Barriers of Physical Activity Participation in Individuals with Intellectual Disability-A Systematic Review. Healthcare, 9(11), 1521. https://doi.org/10.3390/healthcare9111521.

Why did the authors focus on balance and not any other physical ability? 

Since there are validated test batteries for the population, what is the gap in the literature that this article aims to fill?

Has the sample size been calculated?

The Materials and Methods section is very confusing, with the authors mixing up participants, instruments, procedures and statistical analyses, which makes it difficult to read.

Round 2

Reviewer 1 Report

Comments and Suggestions for Authors

I wish to thank all authors for the opportunity of reading their very interesting and informative work. Authors have thoroughly addressed all issues raised in my previous review and that I have no further comments. Therefore, based on my expertise, the paper may be accepted for publication in its current form.

Reviewer 3 Report

Comments and Suggestions for Authors

When I asked for the reference to be updated, I wasn't referring to the format, but to the year of publication. There is already a more recent reference to the one submitted by the authors.

Otherwise, I appreciate the authors' efforts to respond to my comments.
